# Recent Advances in Fabrication Methods for Flexible Antennas in Wearable Devices: State of the Art

**DOI:** 10.3390/s19102312

**Published:** 2019-05-19

**Authors:** Bahare Mohamadzade, Raheel M Hashmi, Roy B. V. B. Simorangkir, Reza Gharaei, Sabih Ur Rehman, Qammer H. Abbasi

**Affiliations:** 1School of Engineering, Macquarie University, Sydney NSW 2109, Australia; raheel.hashmi@mq.edu.au (R.M.H.); roy.simorangkir@mq.edu.au (R.B.V.B.S.); 2Faculty of Engineering, Islamic Azad University-South Tehran Branch, Tehran 15847-43311, Iran; reza.qaraei@gmail.com; 3School of Computing and Mathematics, Charles Sturt University, Port Macquarie NSW 2444, Australia; sarehman@csu.edu.au; 4School of Engineering, University of Glasgow, Glasgow G12 8QQ, UK; Qammer.Abbasi@glasgow.ac.uk

**Keywords:** embroidery, microfluidics, polymers, washable devices, wearable devices

## Abstract

Antennas are a vital component of the wireless body sensor networks devices. A wearable antenna in this system can be used as a communication component or energy harvester. This paper presents a detailed review to recent advances fabrication methods for flexible antennas. Such antennas, for any applications in wireless body sensor networks, have specific considerations such as flexibility, conformability, robustness, and ease of integration, as opposed to conventional antennas. In recent years, intriguing approaches have demonstrated antennas embroidered on fabrics, encapsulated in polymer composites, printed using inkjets on flexible laminates and a 3-D printer and, more interestingly, by injecting liquid metal in microchannels. This article presents an operational perspective of such advanced approaches and beyond, while analyzing the strengths and limitations of each in the microwave as well as millimeter-wave regions. Navigating through recent developments in each area, mechanical and electrical constitutive parameters are reviewed, and finally, some open challenges are presented as well for future research directions.

## 1. Introduction

Recent advancements in wireless technology have led to the advent of wireless body area networks (WBAN) due to their wide range of applications in medical and nonmedical fields. This has drawn broad interest in the field of wearable antennas as the focal point for any WBAN sensory systems. Several frequency bands have been identified for research and commercialization of WBAN communication systems which recently include the millimeter-wave (mmW) bands (see Table 1). 

Common application scenarios in WBANs, as illustrated in Figure 1, include:Inter-node communication between various sensory nodes attached on the surface of the bodyIntra-node communication between a master node on the body and a transceiver off the bodyImplanted sensory nodes communicating with a node on the body (on-body node acting as a relay)Implanted sensory node communicating directly with a transceiver off the body.

Depending on the type of application, it is vital to choose a suitable form for the antenna, as a one-design-fits-all approach often does not meet all requirements. Several forms have been proposed for wearable antennas. Among them are planar monopole antennas [1], integrated inverted-F antennas (IIFA) [2], planar inverted-F antennas (PIFA) [3,4,5], microstrip patch antennas [6], magneto-electric dipole antennas [7], substrate-integrated waveguide (SIW) antennas [8,9], EBG-based antennas [10], dipole antennas [11], and cavity slot monopoles [12] in the lower frequency bands. In addition, Yagi-Uda antennas have been proposed as an appropriate candidate for on-body communications at mm-wave, having a good compromise in terms of size and gain performance [13]. Key considerations for any of these applications include the form of the antenna, electrical stability when placed on the body, robust fabrication process that provides increased mechanical stability to cope with variations of body posture, motion artefacts, subject specificity, and moisture (humidity/perspiration). Another important consideration is the unobtrusiveness of such antennas. The need for unobtrusiveness dictates that the antennas must be integrated within the clothing of the wearer, or be attached to the body, so as to cause minimal or no interference with the usual routine of the individual. This consideration is very important to negate the user inconvenience in daily activities, thus increasing the acceptance of the system in any circumstances.

Considering these requirements, a number of challenges exist pertaining to the fabrication of wearable antennas. Most conventional antennas are fabricated using printed circuit board (PCB) technology based on rigid FR4 substrates and other fiberglass/ceramic materials, for instance those from Rogers or Taconic. The resulting antennas are often obtrusive when deployed on the body, and are difficult to readily integrate within clothing. Secondly, such antennas and RF circuits must be encapsulated in protective casings to prevent oxidation and corrosion resulting from perspiration, hence making them stable across a wide range of environment conditions. The casing is also advantageous for protecting the antenna from mechanical stress resulting from changes in posture. In recent years, advancements in manufacturing techniques have led to several efforts to address the challenges stated above. These efforts can be largely classified into the following categories, based on the type of manufacturing process employed:Fabric-based embroidered antennasPolymer-embedded antennasMicrofluidic antennas with injection alloysInkjet printing, screen printing and photolithography3D-printed antennas

A number of papers are in print providing substantial reviews on antennas for BAN systems [14,15,16,17,18,19,20,21,22,23,24,25,26,27]. In [14,15,16,17,18,19,20,21], the focus of the review has been heavily on the antenna designs, discussing their considerations, challenges, and limitations in relation to the antennas’ operation near the human body. In [17,20,22,23,24,25,26,27], various materials and manufacturing technologies for realizing antennas for wearable applications are reviewed. However, in general, they only focus on a specific class of material or fabrication technique, for instance, textile or inkjet/screen printing. 

In contrast, in this paper we provide a complete survey of recent materials and fabrication methods that have been applied up to now to realize antennas for body area networks, ranging from VHF to millimeter-wave band. They include those utilized for realizing the classes of wearable antennas mentioned above—i.e., fabric-, polymer-embedded, microfluidic, and print-based antennas—as well as the emerging futuristic millimeter-wave wearable antennas. Such a review paper allows a complete view of possible methods to realize antennas for wearable applications.

The rest of the paper is organized as follows. In Section 2, Section 3, Section 4, Section 5 and Section 6, we describe the fabrication process and the materials used for the categories of wearable antennas we mentioned earlier (i.e., embroidered, polymer-embedded, microfluidic, inkjet, screen, and 3D-printed antennas). In each category, discussions on the important considerations and challenges are given. We also provide discussions on the fabrication techniques for future wearable millimeter-wave antennas in Section 7 and human-tissue-equivalent phantoms in Section 8. Finally, the paper concludes in Section 9.

## 2. Fabric-Based Embroidered Antennas

Antennas integrated within clothing provide a convenient alternative to those fabricated using rigid substrates. Such antennas are constructed by embroidering the desired shape of the radiating elements using conductive threads directly on the clothing. As embroidered antennas have seamless integration with clothing, they successfully address the unobtrusiveness requirement of wearable electronics. They also have a robust physical capability to stand repetitive deformations due to the contour changes of a human body.

In addition, the computerized fabrication process leads to faster production which is especially useful in mass production. A computer-aided design (CAD) software controlled sewing machine usually is utilized to make embroidered antennas. From the images of the design, imported to the CAD software, a digitized design is generated and uploaded to the sewing machine to be used as the embroidery guide. A careful adjustment on the yarn tensions, embroidery speed, and stitch gap has to be done before starting the embroidery. However, for a simple patch structure, the conductive thread can also be embroidered by hand.

In general, textiles have an advantageous characteristic of low permittivity due to its porous nature. In [28], the measurement of the dielectric constant of fabric through the resonance method is presented. The dielectric constants of six fabric materials—including jeans cotton, polyester-combined cotton, and polyester—have been determined. As it shows that all of the permittivity values are lower than two.

Antennas developed through this technique can be in a coplanar topology or backed by a ground plane. Figure 2 shows the prototype of an E-shaped embroidered antenna that we designed and fabricated, for a frequency of 5 GHz. The antenna has an embroidered E-shaped microstrip patch on one side and an embroidered ground plane on the other side. This antenna was embroidered by cross-stitching five-filament copper thread from Elektrisola on a polyester stabilizer, and then sewn onto felt material that was used as the substrate.

Key considerations in embroidered antennas include characterizing the material properties of the fabric, as well as the conductive threads, and to a reasonable extent, the precision of the embroidered patterns, the stitch density, and the thickness of the fabric [29]. Various mechanical factors—such as thread tension, embroidery speed, needle type, thread size, and thread type composing the lockstitch pattern—are also found to be crucial, not only for the antenna performance but also for antenna reproducibility [30,31]. Some conductive threads are commercially available, such as X-static [32], Shieldex [33], Elektrisola [34,35], MCEY [36], and Amberstrand [37]. Table 2 shows some examples of embroidered antennas fabricated with various conductive threads and applied on a variety of substrates.

The density of the embroidered patterns must be high enough, to the extent that the distance between adjacent threads become very small compared to the wavelength [38,39]. Antenna parts made out of embroidery, as opposed to conductive fabrics, are by nature inhomogeneous, and their conductivity is dependent on the pattern and the stitching density due to the direction of the current flow (see Figure 3). This in turn can cause a shift in the resonant frequency, and affect the efficiency. Sparse patterns shift the resonant frequency of the antenna towards lower frequencies, resulting from the apparently larger electrical size [29], whereas unnecessarily increasing the stitching density decreases the electrical length by causing local cancellation of the electrical current over parts of the antenna [32]. Comparative studies were conducted for stitching density in [38,40], and the warp and weft directions were identified as the most promising solution in order to obtain reasonably good antenna performance.

In [29], a comparison between three patterns of embroidery: horizontal, vertical, and diagonal lines is presented. The result shows that the vertical thread orientation is the preferred direction for the first radiation mode and gives the best performance for higher-stitch-density antennas. Moreover, diagonal-stitch-direction antennas tend to perform better than horizontal ones due to the current’s flowing in the preferred direction. Moreover, it is shown that the resonant frequencies for horizontal and diagonal patches are lower than those for the vertical stitch direction possibly due to the increased current path lengths and the increased inductance and capacitances caused by the anisotropic nature of the conductors.

It was implied in [29] that the best performance of an embroidered antenna can be achieved if the embroidery is done following the pattern of the antenna current distribution. The problem then comes from antennas having complex current distributions, e.g., wideband antennas or higher-order-mode antennas [39]. Generating a digitized embroidered pattern following the actual current pattern of such antennas is not always an easy process.

However, a study has been done in [39] suggesting that for the case of antennas with complex current distributions, the process can be simplified by using an appropriate simple stitching pattern, for instance, the vertical and horizontal directions. Increased thread density—i.e., the adjacent threads touching each other or the inductive coupling between them being very high—can be applied so that it resembles a continuous conductive surface, although there are unavoidable compromises in the performance and cost.

Some other factors that need to be considered in the embroidering technique are the precision, conductivity, and cost of the embroidery process. In some efforts, precisions down to 0.3 and 0.1 mm—which are typical geometrical precisions of PCB fabrication—have been achieved by utilizing commercial conductive threads with smaller diameters, e.g., 40- and 20-strand Liberator, and 7-filament silver-plated copper Elektrisola [30,34,35]. These threads were found to have a relatively low DC resistance as well. The approach has been successfully applied to generate various design topologies with varying levels of complexity, for instance dipole, archimedean, sinusoidal, toothed, and trapezoidal shapes.

Considering the price of the commercial conductive fabric, the embroidery process was in general done once over the desired pattern, to reduce the overall fabrication cost. However, some researchers—e.g., [41]—did apply the embroidery process on the designed pattern twice, the double-layer embroidery technique. This was done to achieve a high stitching density, low physical discontinuities, and hence a higher conductivity of the embroidered layer.

Moreover, in order to decrease the cost of the embroidering techniques, a non-uniform embroidery technique was presented [42]. In this method, the conductive threads were embroidered non-uniformly across the antenna patch, following a more precise observation of the current distribution of a fundamental-mode rectangular patch. The number of horizontal lines in the uniform-mesh patch antenna (NMPA) is reduced, since the horizontal conductor paths were found to be less important in the TM_01_ mode. An NMPA structure placed on felt was compared with its counterpart having a 100% conductor coverage. The results showed that, with only 20% conductor coverage, an efficiency of 60% can be achieved while the centre frequency and return loss at the 2.45 GHz were not affected. The work suggested that the technique can reduce significantly the cost of manufacturing and allow for a more flexible antenna with a minimized antenna performance reduction.

Some challenges related to the embroidery technique include the degeneration of the conductivity of the embroidered layer over washing which, in turn, affects the antenna properties [45]. In addition, they are prone to fraying during the embroidery process, unless right the needle, proper speed, and tension are applied, which needs a very precise fabrication process. As an alternative to embroidery, commercial woven conductive fabrics which are available in the form of fabric sheet are used to realize the antenna metallic patterns stitched into the clothing, but they suffer from the same issues of degeneration of the conductive layer as well as fraying which may be overcome by shaping fabrics by means of a laser cutting machine (see Figure 4). 

## 3. Polymer Embedded Antennas

Wearable antennas that are made using flexible polymers are an alternative to fabric-based embroidered antennas. Polymers in general are low-permittivity materials with low loss, which make them a suitable option for higher-frequency applications. In addition, they are stable to a wide range of environments due to their low water absorption together with high flexibility and stretchability. For this reason, flexible polymers such as polydimethylsiloxane (PDMS) [46] and liquid crystal polymer (LCP) [47] are used as the antenna substrate.

A major challenge identified in utilizing polymers for antenna development is the poor adhesion of metal to the polymer, which might cause separation of the antenna conductive parts from the non-conductive parts such as the substrate. For this reason, polymers were employed to construct wire-type antennas, which require minimal surface contact between the conductive part and the polymer. Some promising solutions to circumvent this challenge embed the conducting parts of the antenna within a thin layer of polymer. This embedding approach has been demonstrated with various types of conductive materials. Among them are carbon nanotube sheets [48], copper mesh [46], silver nanowires [49,50], embroidered threads [51], and—recently—conductive fabrics [52,53] which turned out to be a simple yet effective solution.

An embedded antenna protects the conductive layer from harsh environments, temperatures, wetness which may be happen from repeated sweating and washing that can affect the weak attachment between the metal and the polymer [48,50]. 

Among polymers, PDMS presents advantages of high flexibility and stretchability and has a reasonably acceptable loss factor at microwave frequencies. The fine-tuning of the PDMS relative permittivity can be easily done through inclusion of other materials such as ceramic powder. The further dielectric losses, lower fringing fields due to the high permittivity which degrade the efficiency of the antenna should be considered [54]. Moreover, a larger amount of ceramic loading affects the flexibility of PDMS and complicates its fabrication process, as the mixture becomes thicker and thus harder to mix homogeneously [55]. The preparation of PDMS can also be done at room temperature through a simple low-cost process [52]. With its characteristics of water resistance, chemical inertness, high heat resistance (up to 400 °C), and stability under ultraviolet exposure, PDMS is very resilient to extreme environments. Table 3 shows several antennas constructed using PDMS. In this table the relative permittivity values range from 2.67–3.8, whereas the loss factors ranges from 0.01 to 0.04. One may note that the relative permittivity and loss factor of PDMS can vary, e.g., for the same frequency of 5880 MHz, an order of magnitude difference in loss factor between the PDMS used in [56] and [57]. Therefore, the characterizing electrical properties of PDMS as part of the antenna design cycle is critical, rather than solely relying on reported data. Various effective approaches for characterizing electrical properties have been reported in [58].

The PDMS-embedded conductive-fabric technique presents excellent opportunities for realization of various types of conformal antennas. The antennas were realized using conductive fabric which later was embedded within the polymer. A rather strong adhesion of conductors to dielectric was achieved. Excellent results in terms of integration can be achieved when conductive fabric is utilized for constructing the metallic parts. The reason is the porous nature of the conductive fabric that allows for the creation of the polymerpolymer bonding. This bonding helps in sealing the PDMS fabric attachment, thus leading to a robust mechanical integration between the two materials. Such a phenomenon was understood though sample observation under a scanning electron microscope (SEM) as shown in Figure 5.

The fabrication process of such polymer-embedded conductive fabric antennas is shown in Figure 6. Antennas can be fabricated through a layer-by-layer process, starting from the bottom to the top layer. Here, PDMS polymer, which is one of the most commonly used polymers for realization of flexible wearable antennas, to explain the fabrication process. 

A step-by-step process for fabricating conductive fabric polymer-embedded antennas, as outlined in [52], is given below:Prepare the PDMS solution with the required permittivity and loss tangent, mold, base, and cut the conductive fabric manually following the dimensions of the antenna.Attach the first mold to the base with a silicone sealant, pour the PDMS solution into the mold, degas in a vacuum desiccator for approximately 30 min, and then cure in an oven at 65 °C for about 2 h.Attach the ground plane on top of the cured bottom encapsulation layer with a thin uncured PDMS solution, repel the bubbles occurring between them, and cure in the oven at 75 °C 20 min.Repeat Step 2 with the second mold to make the middle PDMS layer.Repeat Step 3 for the main patch.Repeat Step 2 with the third mold to make the top encapsulation layer.Carefully peel the prototype from the mold. If necessary, trim the excess PDMS layers from the edges of the antenna.

Figure 7 shows the top, side and bending view of two microstrip antenna prototypes fabricated based on conductive fabric embedding technique on two substrates: pure PDMS and PDMS-ceramic composite [60].

In the embedding technique, the conductivity of the conductive parts is relatively low compared to the metals which influence the antenna performance. In addition, the loss tangents of the polymers—especially in the higher frequency—is high, which influences the antenna efficiency and as a result affects the antenna’s gain. To compensate for this, it was suggested in [60] to use different conductive fabrics for different conductive parts of the antenna, to double-coat the fabric, to develop a coating material with higher conductivity, or to decrease the loss of the polymer through other-material inclusions.

## 4. Microfluidic Antennas with Injection Alloys

One of the interesting methods to fabricate flexible antennas such as dipole antennas is injecting metal alloys. In this method, liquid metals and liquid–metal alloys such as eutectic gallium 75% indium 25% (EGaIn) and Galinstan can be utilized as flexible conductive parts and will be injected into the flexible substrate through a series of meshed micro-fluidic channels. For the construction of the microchannels, silicone TC5005 and polydimethylsiloxane PDMS are commonly employed. Table 4 shows several example antennas where PDMS was used to construct the microchannels. This method provides an alternative to the use of conductive sheets on a flexible substrate, thus providing extreme resilience towards mechanical stress and duress, alongside high conductivity, as shown in Table 4. The use of injection alloys yields an increased antenna efficiency due to the higher conductivity than commercial conductive fabrics [61]. Table 4 shows that the radiation efficiency of this method is mostly high, at more than 70%, even under deformation and bending.

With advanced developments in the field of microfluidics, very fine microchannels can be constructed, leading to high precision, miniaturized flexible antennas. These antennas’ stretchability is in general high. In [62], the antenna is stretchable by a strain of up to 300% and the electrical length of the patch antenna varies with the stretching, demonstrating a frequency tuning capability from 1.3 to 3 GHz. The injection of metal in microchannels can be controlled through a voltage bias, as well as by controlling the pressure within the channels. Moreover, antennas based on injection alloys and with microchannels that can self-heal in response to sharp cuts were reported in [61].

So far, several monopole and dipole antennas and antennas with co-planar ground planes have been designed using this method [61,62,63,64]. Wire-type antennas have been found to be more robust with this technique, as microchannels with relatively small diameters allow for an even distribution of the metal along the channel. Wider microchannels have been identified as more difficult to obtain and to maintain a uniform distribution of metal. Also, it is often difficult to shape the liquid metal into channels with a low aspect ratio of height and width. Printed antennas, such as slot or microstrip patches, are challenging to realize with this technique.

Significant advances have been made in microfluidics and therefore, various approaches to inject liquid metal have been reported. A handmade patch antenna fed through an aperture coupled slot was constructed using a planar reservoir of Galinstan (an alloy of Ga, In, and Sn) by inserting it in a TC5005 silicone substrate [60]. Galinstan was injected into the reservoir using a syringe in this case. In [65], microchannels are employed to shape the patch, with the ground plane at the back of the antenna (see Figure 8).

Taking advantage of the rheological properties of the metal, 1 mm-wide microfluidic channels were used in [64] to construct a flexible microstrip patch antenna operating at 3.4 GHz. The experimental characterization of this antenna demonstrated a radiation efficiency of 60%, and stable performance at various curvatures was observed. Some more versatile antennas have also been made using this approach. Figure 9 shows the microfluidic channels which are filled with the EGaIn for a microstrip antenna.

In [63], a flexible planar inverted-cone antenna (PICA) which was fabricated by injecting room-temperature liquid metal alloy into micro-structured channels in a sheet of PDMS, is presented. The presented PICA operates at 3 to 10 GHZ with a radiation efficiency of more than 70%. In [65], a three-dimensional flexible IFA operating at 885 MHz on NinjaFlex flexible plastic was demonstrated, made by using Galinstan liquid metal.

An important consideration in applying this fabrication technique is to ensure that air does not get trapped in the microchannels during the liquid-metal injection process, as the resulting discontinuities would disrupt the flow of currents, and thus the functionality of the antennas. This is in addition to the possibility of fatigue or the liquid metal leaking from the flexible substrate under harsh environments or high stress circumstances.

## 5. Inkjet Printing, Screen Printing, and Photolithography

Printing on a flexible substrate is another attractive approach for realizing lightweight and flexible antennas due to its possibility to be applied to various types of substrates. Various types of antennas with different ranges of operating frequency have been successfully developed by this technique as can be seen in Table 5, although, coplanar waveguide (CPW) based antennas are normally preferred owing to having the conductive parts (e.g., ground plane and radiator) on one side, and hence reduce the fabrication complexity.

Inkjet printing as one of the printing technologies has been emerging as a popular solution for a fast antenna prototyping technique with relatively low cost [72,73]. Due to the utilization of ink droplets of a size down to a few picoliters, the printer can produce a pattern with extremely high resolution [74]. In addition, the use of a clean room is not compulsory. For the conductive materials, silver, carbon copper, and gold nanoparticles-based conductive inks have been extensively used. On the other hand, for the substrates, the use of various types of flexible materials have been successfully demonstrated, including polyethylene naphthalate (PEN) [73], polyethylene terephthalate (PET) [75], [76] kapton polyimide [77], paper [78], polyester cotton [79], and PDMS [80]. Figure 10 shows an inkjet-printed patch antenna prototype depict an impedance bandwidth of 26–40 GHz on PET substrate [81].

Some features that determine the quality of the printing in the types of drop-on-demand (DoD) and continuous inkjet: include the ink properties (e.g., viscosity and surface tension), surface characteristics of the substrate (e.g., wettability), layers of printing for conductivity, the platform temperature, the print head parameters and the particle diameter. In addition, in fabrication, some concerns such as nozzles clogging and incompatibility of some types of conductive inks due to larger particle size should also be considered.

An example of a fabricated antenna using the inkjet printing method has been provided in [77], where silver nanoparticle ink was utilized on a Kapton substrate by using a Dimatix DMP-2831 printer. In this work, the 30 µm drop spacing of the nano-particles and the five conductive layers were carefully chosen to obtain accurate dimensions as well as good conductivity.

It is still challenging to inkjet print on rough, uneven, or porous textiles due to the low thickness of the conducting layers, and the inability of the textile to withstand the ink curing temperature and bending. Some efforts try to address these issues. In [72], by developing an interface coated layer which bonds to a standard polyester cotton fabric, a smooth surface is created for inkjet printing. Antenna efficiencies of greater than 60% have been achieved with one layer of conducting ink, which also saves a considerable time and cost of fabrication and improves the printing resolution. The proposed antenna silver (U5714) is chosen for the conductive ink. The authors in [82] also introduce a screen-printed interface layer to reduce the surface roughness of the polyester/cotton material that facilitates the printing of a continuous conducting surface. The proposed antenna is placed on Taconic RF-45, and an efficiency of 53% was achieved with two layers of ink.

Photolithography is another printing technique, and is a process of producing metallic patterns using photoresist. Complex patterns can be produced by this method but important matters such as the conductivity, involvement of hazardous chemicals, and the length of multistep process must be considered [66]. One photolithography technique is line patterning, which involves the design of a negative image of the desired pattern using a computer-aided design program. This process is followed by depositing a conductive polymer on the substrate/film, and then taking out the printed mask by applying ultrasonic energy to the substrate [66].

## 6. 3D-Printed Antennas

The popularity of the additive 3-D printing technique for wearable and conformal antennas is because of its extensive advantages: commercial availability of printers and materials, in-house-made, fast fabrication process, capability to fabricate complex 3-D structures with multiple materials and capability to change the density of the printed object [83,84,85,86]. The introduction of NinjaFlex filament as a highly flexible and stretchable material from Fenner Drives, Inc. in 2014 enabled 3D printing to be applied to wearable antennas. Among current 3-D manufacturing technologies—fused deposition modeling (FDM), stereo-lithography (SL), and laser sintering—the FDM technique allows printing on patch patterns cut from mainly plastic-based materials [84]. A number of methods are presented to fabricate the flexible antenna with the 3-D printing manufacturing technique. In [84], a square patch antenna at 2.4 GHz is fabricated with a novel 3-D printable flexible filament, based on NinjaFlex, and is tested under various bending conditions. This novel 3-D printable flexible filament is presented for manufacturing the substrate of a patch antenna. The printed material density is changed to control the dielectric constant and thickness of the substrate. A loop antenna at 2.4 GHz is also fabricated by 3-D printing technology with 3-D printer based on FDM and using NinjaFlex, a flexible 3-D printable material, as a 3-D hemisphere substrate [87]. An ultra-wideband (UWB) monopole antenna on an additive manufactured flexible substrate based on fuse-filament fabrication technology is proposed. In [88], graphene-based RFID tags on a 100% cotton fabric by using a water-based graphene ink (HDPlas IGSC02002) is fabricated with an nScrypt tabletop series 3D direct-write dispensing system. The aim of the work was checking the possibility of applying 3D direct-write dispensing to form the antenna–IC connection. In [67], the fabrication of the antenna is done using brush painting. The stretchable silver conductive paste is brush-painted on the NinjaFlex 3-D printed substrate, and the structure is thermally cured.

## 7. Fabrication Techniques for Future Wearable Millimeter-Wave Antennas

In light of the advanced manufacturing techniques, we have reviewed so far the frequency band up to UWB. In this section, the progress and challenges in developing wearable antennas at millimeter-wave frequency bands are discussed.

MmW technology has been given much attention recently because of the huge bandwidth potential that is available. Even small bandwidths like 5–10% at microwave bands result in reasonably large bandwidths at mmW, which in turn can be used for ultra-fast data transmission [89]. Such a characteristic is very useful for latency-intolerant applications such as those in implantable communications service and robotic invasive procedures in healthcare. Thus, wearable antennas at mmW are attractive for short-to-medium range high-performance applications.

Characterization and modeling the propagation in the human-body environment is paramount, as this information is essential, not only for preliminary design of wearable antennas and systems, but also for in vitro testing of such antennas, considering the regulatory issues. Human-body features become electrically large at mmW frequencies, particularly compared to the very small and fine features of antennas, and therefore classic full-wave techniques become inefficient.

Potential numerical approaches and associated trade-offs with methods such as ray-tracing model, finite difference time domain (FDTD), and the method of moments (MTM) are discussed in [90].

Some studies on-body propagation at 60 GHz analytically, numerically, and experimentally by using a skin-equivalent phantom are done in [91,92,93]. In [92], propagation along a flat skin-equivalent phantom at 60 GHz in the presence of textiles using a Green’s function is investigated. The effects of the body on a textile antenna shows a decrease of the path gain by 2–5 dB. These results show an increase of the textile thickness in the 0.2–2.6 mm range results in an increase of the path gain. In addition, it is shown that the power decay exponent is not significantly affected by the presence of a textile. In [93], it is shown that, the antenna impedance matching and gain are almost unaffected by the presence of the body even when the distance between the antenna and the body is small. The measured specific absorption rate (SAR) for a considered antenna also shows that the exposure level of the body is below the exposure limits established by the International Commission of Nonionizing Radiation Protection (ICNIRP) when the input power is lower than 550 mW for an antenna and a body gap of 1 mm. Antennas fabricated using textiles and inkjet printing have been successfully reported at mmW frequencies, with reasonable performance and efficiency. A 2 × 2 microstrip patch antenna over a cotton-woven fabric substrate array at 60 GHz was proposed in [94]. The fabric was characterized to find a relative permittivity of 2.0 and a loss factor of 0.02.

Dictated by the requirements for fine feature sizes, 0.07 mm thick copper foils are used for radiating elements in the array, and the feed network has a minimum line width of 0.4 mm with a tolerance of ±0.15 mm. The insertion loss of a 50 Ω microstrip line fabricated with this method, printed on a 0.2 mm thick textile substrate, was measured to be 1.6 dB/cm.

The antenna elements were micromachined using a laser, providing a geometrical accuracy of approximately 10 μm. The array demonstrated large bandwidth, from 57–64 GHz.

Inkjet-printed Yagi-Uda antennas were reported in [95] at 24.5 GHz. With the Dimatix DMP-2831 inkjet printing system, Cabot CCI-300 silver nanoparticle ink was used to print metallic patterns on a flexible liquid-crystalline polymer (LCP) laminate. A difference of approximately 2 dB in realized gain is observed at the center frequency, spreading out gradually over the rest of the band from 23–26 GHz.

This difference can be attributed to the complexity of the multi-layer printing process, which is critical to ensure a uniform substrate thickness, and the fineness of the edges of the metallic patterns. A similar approach with a fabric substrate (ε_r_ = 2.5, **tan**
***δ*** = 0.016) was used to demonstrate a printed Yagi-Uda antenna at 60 GHz, for the along-the body propagation mode [14]. To achieve high feature-size accuracy, copper foil was glued to the fabric substrate, and the desired features were cut using a laser machine directly. A full-ground plane is used on the other side of the fabric, made out of the same copper foil. On-phantom characterization was carried out for this Yagi-Uda antenna, showing a gain of 9.2 dBi and an efficiency of 48%. In [96], an array antenna at the frequency of 26.5–40 GHz based on inkjet-printing with a 0.5 μm thin silver pattern with a resolution of less than 20 μm on an LCP substrate is presented. Figure 11 shows the proposed fabricated mmW antenna and two-element mmW array antenna [96].

Polymer-based antennas using PDMS are relatively challenging in this case. A thin PDMS membrane was used to construct a 4 × 2 microstrip patch antenna array. A 50 Ω microstrip line was used to characterize PDMS for V-band, with ε_r_ = 2.68 and tan *δ* = 0.04 [97]. Measurements showed an insertion loss of nearly 3 dB/cm, which presents a significant limitation to the transmission-line length and the required scalability of the feed network. The antenna array demonstrated wideband performance in V-band (50-60 GHz), with a measured gain of 12.3 dBi and a radiation efficiency of 28%.

Despite the challenges, PDMS has its advantages for mmW applications, especially due to its compatibility with microactuators and MEMS, where antenna integration can be improved to avoid long transmission lines, thus circumventing efficiency degradation.

## 8. Human-Tissue-Equivalent Phantoms

The unique operating environment of wearable antennas, i.e., near human body, impose additional requirements in the design phase of the antennas. The electromagnetic coupling between the antenna and the lossy human body tissue affects the antenna performance including the matching, resonance frequency, efficiency, and gain [98,99]. Concurrently, the antenna radiation brings adverse impact to the human body tissue. Therefore, the existence of human body has to be taken into consideration during the design simulation and most importantly during the experimental validations of the wearable antenna. In the simulation stage, it is a common practice to simulate the antenna in a close proximity to homogeneous or heterogeneous phantom models. This is then validated by measuring the antenna on the body of a human test subject or fabricated homogeneous or heterogeneous phantoms emulating the electrical properties of specific human body tissues. One of the challenges in phantom development for wearable antenna testing is the fact that human tissues are complex multilayer materials having a wide range frequency dependent electrical properties. Moreover, these properties vary with age and are different from one individual to another [100,101,102,103,104].

There have been significant efforts made on investigating the development of phantom that emulates the electrical properties of specific body tissues in certain range of frequencies [105,106,107,108]. A comprehensive review on phantom development can be found in [108], which elaborate the different categories of phantom, their advantages and disadvantages, the applications as well as the fabrication process in a systematic way.

Depending on the water content of the phantom, it generally can be classified into two major groups, the low-water- and high-water-content phantoms. Based on the physical structure of the phantom, it also can be divided into four major categories, i.e., liquid, semiliquid/gel, semisolid/jelly, and solid phantoms. Introducing new flexible materials for characterization of antennas-under-test (AUT) to emulate environments like human body parts (e.g., arms, torso, etc) or the head, including the skull and the brain.

The first three classes of phantom have water as their main constituent. Consequently, they are suitable for human body tissues with higher permittivity, such as muscle and brain. The preparation of liquid phantoms is relatively the easiest as it involves only the mixing of composing materials following an appropriate recipe (see [109]). The challenge with such phantom is that it suffers from the inconsistent electrical properties—i.e., relative permittivity and conductivity—as a result of dehydration and materials spoiling due to fungi contamination.

Semi-liquid phantoms are slightly more solid than liquid phantom although are still in need of container to hold the shape. This class of phantom are also found to be more consistent over time in terms of the electrical properties. With even more solid physical structure, there is a class of semi-solid phantoms which can hold their shapes independently [110]. This characteristic makes this class a suitable option for emulating realistic operating environment of the antennas, i.e., soft and multilayered body tissues. The phantoms can also sustain the electrical properties over time better. However, unlike the first two classes of phantom, the semi-solid phantoms are usually not adjustable, and hence not reusable.

In the solid or dry phantoms [106], the main material is ceramic materials, which are available in wide range of a wide permittivity with relatively low loss. The latter specifically brings the challenge to emulate the actual loss tangent and conductivity of many types of human body tissue. Another issue is with the adhesive materials which are added during the fabrication process to remove the air gaps between the adjacent pieces of the ceramic phantoms. These inclusions of these adhesive materials make the phantoms hard to cut or reshape. To do invasive measurements with this class of phantom is indeed problematic considering its physical structure.

However, solid phantoms have the advantages of shape durability and electrical properties sustainability over a very long period of time. In [111], a dynamic solid phantom is presented. This phantom is developed with the help of mannequin, the linear motion actuator and the skin-equivalent, to evaluate the performance of the doppler radar for the breath detection [112].

## 9. Conclusions and Future Research Directions

In this paper, all of the current advanced manufacturing methods for wearable antennas in the categories of fabric based embroidered antennas, polymer-embedded antennas, microfluidic antennas with injection alloys, and inkjet screen and 3D-printed antennas, are discussed. Every method is discussed in detail and all of the considerations regarding the fabrication process, materials used, and antenna properties are mentioned. For any presented technique, several examples as a reference are provided. In the tables, properties of the used material, the frequency, and antenna types are reviewed. In conclusion, all of the manufacturing techniques and their properties are summarized in Table 6 to facilitate the comparison of the mentioned techniques and help choose a proper method regarding for the antenna application. 

With the growing interest in wearable antennas, especially for WBAN applications, some of the most important open research topics and predicted future research in this domain are given as follows:Development of methods for human-tissue parameters extraction and body models at mm-wave and terahertz frequencies in order to develop accurate and practical mm-wave antennas.Improving the efficiency and precision of the current manufacturing techniques, especially in higher-frequency bands.Improving methods of fabrication for microchannels in antennas based on injecting alloys.Introducing new flexible materials as a substrate with a wider range of permittivity for the embedding technique.Introducing new conductive fabrics and threads in the market with higher conductivity or less resistivity.Introducing new flexible materials such as graphite films [113] and graphene film (FGF) [114] as a conductive parts of antenna with a high conductivity for printing technique or new proposed fabrication techniques.

## Figures and Tables

**Figure 1 sensors-19-02312-f001:**
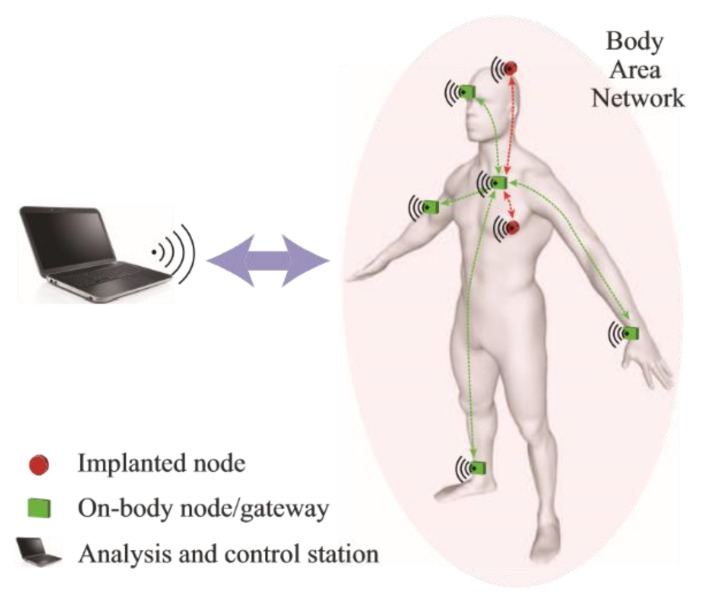
Illustration of wireless body-area network.

**Figure 2 sensors-19-02312-f002:**
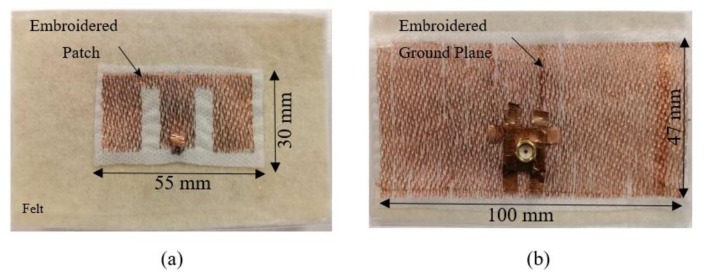
E-shape antenna fabricated based on embroidering technique (**a**) top view; (**b**); bottom view.

**Figure 3 sensors-19-02312-f003:**
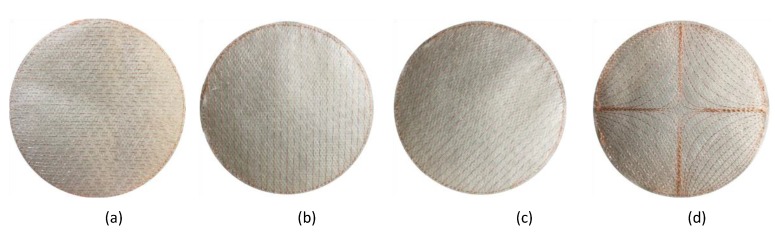
Embroidered patch structures with different stitch directions (**a**) horizontal pattern; (**b**) vertical pattern; (**c**) diagonal pattern; (**d**) complex pattern.

**Figure 4 sensors-19-02312-f004:**
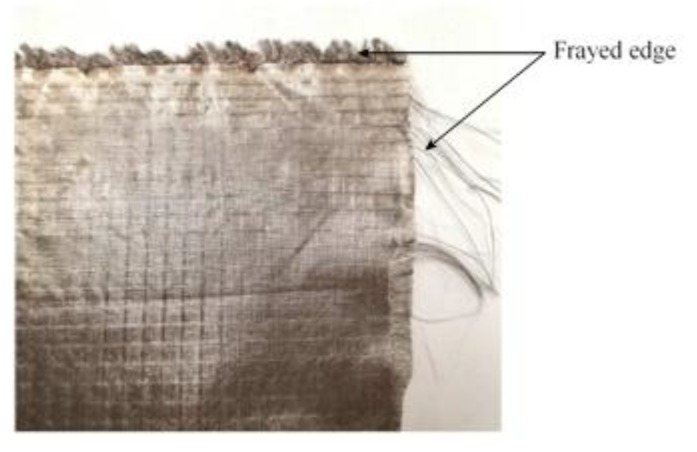
Fraying in the edge of the fabrics.

**Figure 5 sensors-19-02312-f005:**
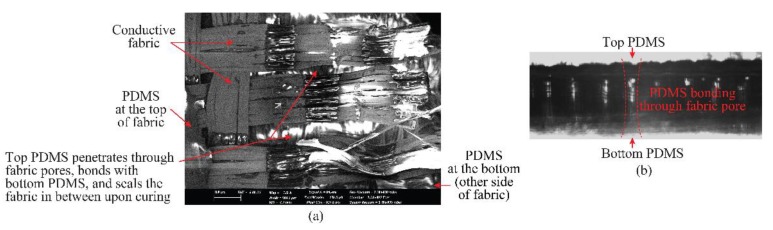
Scanning electron microscopy of the fabric (**a**) top view; (**b**) side view.

**Figure 6 sensors-19-02312-f006:**
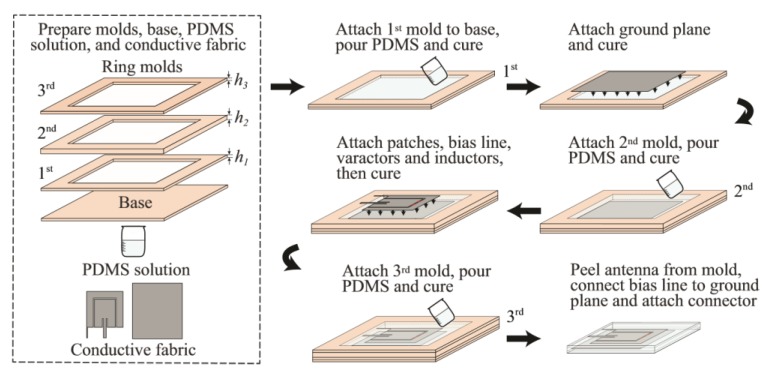
Manufacturing process flow for embedding fabrics in PDMS.

**Figure 7 sensors-19-02312-f007:**
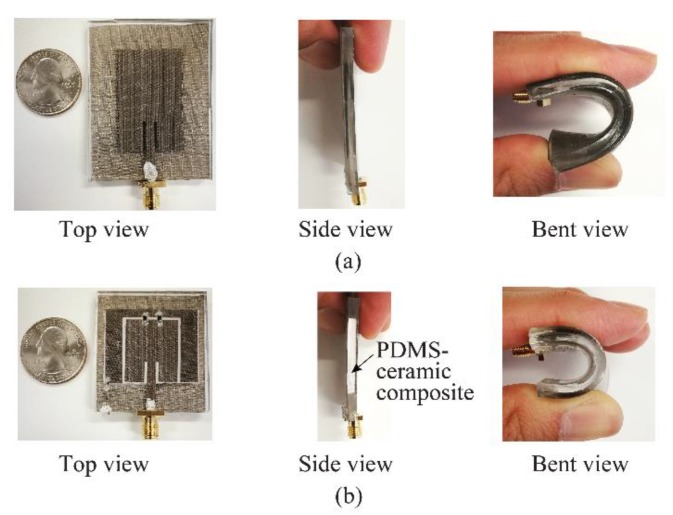
Photographs of the fabricated PDMS-embedded conductive fabric antennas (**a**) with pure PDMS; (**b**) with PDMS-ceramic composite [60].

**Figure 8 sensors-19-02312-f008:**
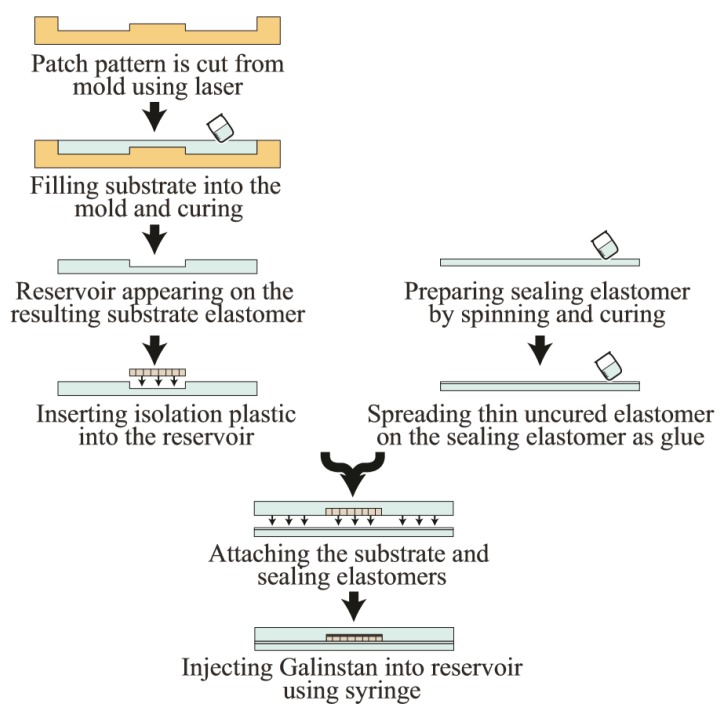
Fabrication steps of the prototype of a handmade patch with injection alloy.

**Figure 9 sensors-19-02312-f009:**
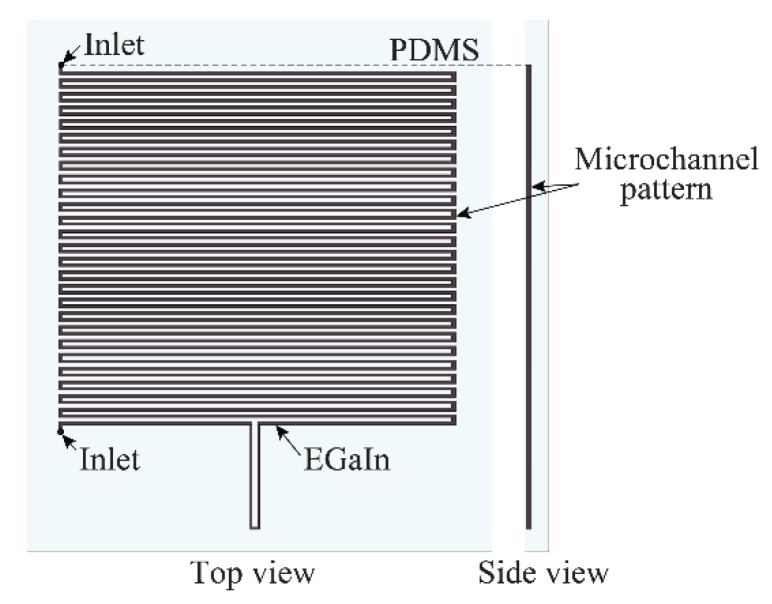
Top view and side view of the microstrip antenna fabricated by injection alloys in the microfluidic channels.

**Figure 10 sensors-19-02312-f010:**
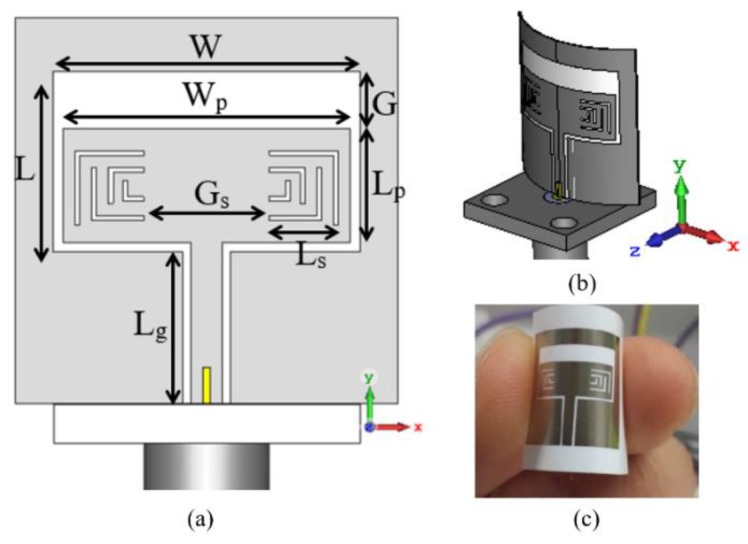
Patch antenna fabricated on PET film based on inkjet-printing technique: (**a**) simulated model; (**b**) simulated bended antenna; (**c**) fabricated prototype [81].

**Figure 11 sensors-19-02312-f011:**
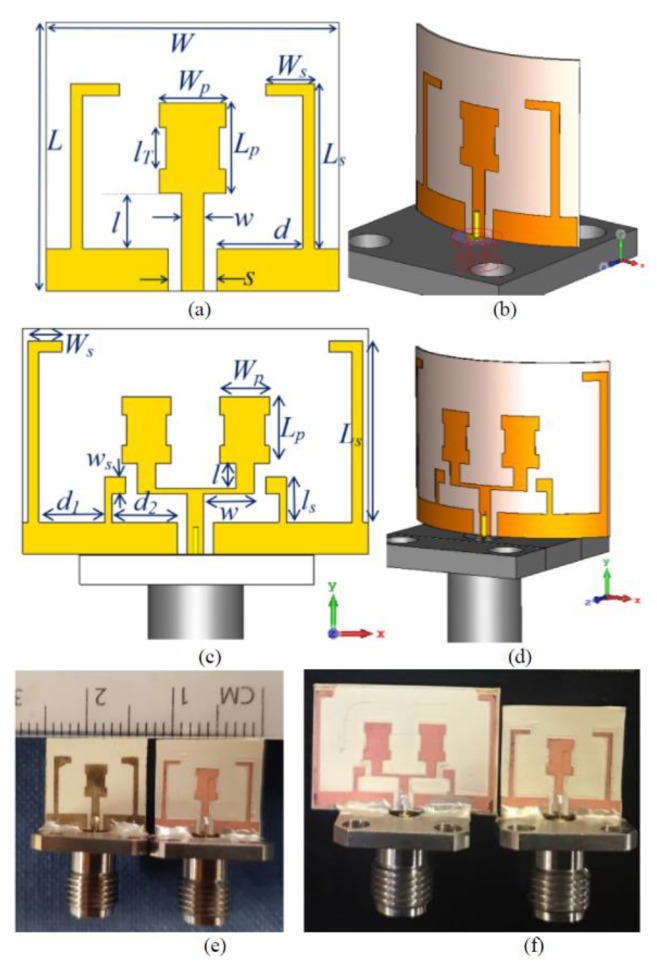
Mm-wave antenna and array fabricated with laser-milling and inkjet printing on LCP substrate. (**a**) Antenna model. (**b**) Conformal antenna model. (**c**) Antenna array model. (**d**) Conformal antenna array model. (**e**) Laser-etched (right) and inkjet-printed (left) fabricated prototype. (**f**) Fabricated antenna (right) and two-element array (left) [96].

**Table 1 sensors-19-02312-t001:** Frequency bands for BAN application.

Frequency	Application
401–402 MHz 402–405 MHz 403.5–403.8 MHz 405–406 MHz	Medical Implant Communication Services (MICS)
413–457 MHz	Medradio Micropower Networks (MMN), transmit and relay data for implanted and body-worn medical devices for diagnostic and therapeutic functions
2.4 & 5 GHz	Wi-Fi, smart hospital beds, mobile nursing stations
2.404–2.478 GHz	Bluetooth, indoor navigation for patients, connectivity between device and smartphone for health-data monitoring
2.36–2.4 GHz	New Medical BAN (MBAN)
New Medical BAN (MBAN)	Industrial, Scientific, and Medical (ISM)
3.1–10.76 GHz	Ultra-Wideband (UWB)
57–64 GHz 59–66 GHz	The band plans and rules at mmW-60 WBAN

**Table 2 sensors-19-02312-t002:** Examples for embroidered antennas.

Ref.	Conductor	Conductor $R_{s}\;$	Substrate Material	Substrate $\varepsilon_{r}$	Substrate tan *δ*	Freq. (MHz)	Antenna Type
[36]	Metal Composite Embroidery Yarn (MCEY)	3.88 Ω/m	Polyester woven	1.15	NA	87.5–108	Folded dipole
[37]	Silver-coated Amberstrand fibers	0.75 Ω/m	NA	NA	NA	900 (41.6%), 1900 (75.8%), 2450 (65.6%)	Patch
[41]	Silver-coated Amberstrand fiber	0.8 Ω/m	PDMS	4.2	0.01	2200	Patch
[42]	Amberstrand Silver 66	6.5 Ω/m	Taconic RF-45	4.5	0.0037	2400	Patch
[43]	Shieldex thread	NA	Cotton fabric	1.8	0.018	900	Patch
[44]	Elektrisola	1.9 Ω/m	Organza	close to air	close to air	760 (15%)	Dipole

Note: $R_{s}$ = sheet resistance; $\varepsilon_{r}$ = permittivity; tan *δ* = loss factor.

**Table 3 sensors-19-02312-t003:** Examples for embedded antennas.

Ref.	Conductor	Conductor *σ or R_s_*	Substrate Material	Substrate $\varepsilon_{r}$	Substrate tan *δ*	Freq. (MHz)	Antenna Type
[48]	Carbon nanotube	0.9 Ω/sq	PDMS	3.8	0.015	2250	Patch
[49]	Silver nanowires	8.13 × 10^3^ S/m	PDMS	2.67	0.01	2920	Patch
[51]	Amberstrand fibers	1.6–2.6 Ω/m	PDMS	3	0.02	915	Wire antenna
[56]	Copper foil	NA	PDMS	2.7	0.013	5800	Microstrip array
[57]	Nickel-copper-silver coated nylon	0.01 Ω/sq	PDMS	2.8	0.02, 0.04	2450 5800	Patch
[59]	Copper micromesh	3.28 × 10^4^ S/m	PDMS	2.8	0.02	2460 to 2940	Reconfigurable patch

Note: *σ* = effective conductivity; $R_{s}$ = sheet resistance; $\varepsilon_{r}$ = permittivity; tan *δ* = loss factor.

**Table 4 sensors-19-02312-t004:** Examples for antennas based on injection alloys.

Ref.	Conductor	Conductor *σ*	Substrate Material	Substrate $\varepsilon_{r}$	Substrate tan *δ*	Freq. (GHz)	Eff (%)	Antenna Type
[61]	EGaIn	3.46 × 10^4^ S/m	PDMS	2.67	0.0375	1.91–1.99	90	Dipole
[62]	Galinstan	3.46 × 10^6^ S/m	TC5005	2.8–3.1	NA	1.3–3	80	Reconfigurable patch
[63]	Galinstan	3.46 × 10^6^ S/m	PDMS	3	0.01	3.1–10.6	70	Planar inverted cone
[64]	EGaIn	3.46 × 10^4^ S/m	PDMS	3	0.05	3.45	60	Patch

Note: *σ* = effective conductivity; $R_{s}$ = sheet resistance; $\varepsilon_{r}$ = permittivity; tan *δ* = loss factor.

**Table 5 sensors-19-02312-t005:** Examples for antenna based on printing technique.

Ref.	Conductor	Conductor *σ or R_s_*	Substrate Material	Substrate $\varepsilon_{r}$	Substrate tan *δ*	Freq. (GHz)	Antenna Type
[66]	Silver nanoparticle	1 × 10^7^ S/m	Kapton	3.5	NA	3.1–10.6	Inkjet-printed monopole
[67]	Silver paste	1.7 × 10^4^ S/m	NinjaFlex	2.8	0.05	2.13–3.25	Patch antenna
[68]	Electrodag	0.025 Ω/sq	Cotton-polyester	1.6	0.02	2.45	Screen-printed patch
[69]	Graphene	4.47 × 10^4^ S/m	3D printed porous elastomer	3.6	0.06	2.45	Inkjet-printed dipole
[70]	Copper	NA	Liquid crystal polymer (LCP)	3.1	0.002	3.1–10.6	tapered slot antenna
[71]	Silver nanoparticle	0.3 Ω/sq	Resin-coated paper and polycarbonate	2.56	0.01	0.8–1.1	Inkjet-printed monopole with AMC on paper

Note: *σ* = effective conductivity; $R_{s}$ = sheet resistance; $\varepsilon_{r}$ = permittivity; tan *δ* = loss factor.

**Table 6 sensors-19-02312-t006:** Comparison of the properties of the presented advanced manufacturing techniques for wearable antenna.

Method	Robustness to Bending	Robustness to Wetness	Cost to Fabrication	Fabrication Simplicity	Weight	Frequency
Fabric-based embroidered antennas	Medium	Medium	Low	High	Low	87.5–5000 MHz 57–64 GHz
Polymer-embedded antennas	High	High	Medium	Medium	High	383–5880 MHz
Microfluidic antennas with alloys injection	High	High	High	Low	Medium	1.3–10 GHz
Inkjet-printed antennas	Medium	Low	Low	Low	Low	1.71–10.6 GHz 23–26 GHz
Fabric-based embroidered antennas	Medium	Medium	Low	High	Low	87.5–5000 MHz 57–64 GHz
Polymer-embedded antennas	High	High	Medium	Medium	High	383–5880 MHz

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
