# Peer review of "Recent Advances in Fabrication Methods for Flexible Antennas in Wearable Devices: State of the Art"

_sensors, 2019, doi:10.3390/s19102312_

Round 1

Reviewer 1 Report

Title: Recent advances in fabrication methods for flexible antennas in wearable devices: State of the art

Comment: The authors summarized a detailed review of various recent materials and fabrication methods for flexible antennas. Overall, the authors introduced in detail only the fabrication methods of different types of antennas. How about briefly introducing the results of the FOM for each type of antenna in Table 6 or conclusions? Though this paper is well-summarized and arranged, some parts should be clarified first:

1.      In Fig. 2, please mark the dimension in the figure for readers to easily estimate the sizes of the embroidered antenna or mention the dimensions of the antenna design. During soldering or attaching the SMA connector with a signal line, were there any delamination or issues?

2.      In lines 198 to 204, the NMPA antenna structure was mentioned with condition coverage only. How about the RF response such as the difference of the VSWR?

3.      The author mentioned the fraying problem in an embroidery antenna structure. If the authors could briefly comment on ways to overcome this problem, it would be beneficial to future readers.

4.      The polymer embedded antenna with PDMS fabric attachment is illustrated in Fig 5. However, it is not easy to understand the sealing effect. The authors can provide more information in Fig 5 to facilitate understanding.

5.      In line 251, the authors mentioned the fine-tuning of the PDMS relative permittivity. Are there any side effects of inclusion of other materials for tuned the relative permittivity?

6.      In line 254, please fix the degree unit (up to 400_C)

7.      If the authors added some images of the microfluidic antennas with injection alloys, it would facilitate better understanding of the concept.

8.      In line 344, why is there a 4. coming out suddenly? Is it just a typo or was something omitted? It looks like the start of another section.

9.      The authors can provide an example or images of the printing technique antennas on page 11 and fabrication techniques for future wearable millimeter-wave antennas on page 12-13.

10.   There are numbering issues after 5. 3D-Printed Antenna followed by 7 — fabrication techniques for future wearable millimeter-wave antennas.

11.   In line 441, a 2_2 microstrip patch antenna was mentioned. Is it correct with 2_2?

12.   The authors mentioned the SAR on page 13-14. There are two main human SAR measurements with body SAR and head SAR. The authors only mentioned the body SAR in the various flexible antennas. What about the head SAR? Are there any technical approaches to reduce the SARs--specific design concept or types of antennas -- without degrading its performance?

Author Response

Comments from Reviewer 1

1.        In Fig. 2, please mark the dimension in the figure for readers to easily estimate the sizes of the embroidered antenna or mention the dimensions of the antenna design. During soldering or attaching the SMA connector with a signal line, were there any delamination or issues?

Authors’ Response: Thanks for the comment. We added the dimension of both patch and ground plane on the Fig. 2. In this prototype, the SMA connector was attached to the antenna as a probe feed. Copper tape was used for this attachment during the measurements without any specific issue, however, conductive glue can be used in place of copper tape as a more robust solution.

2.        In lines 198 to 204, the NMPA antenna structure was mentioned with condition coverage only. How about the RF response such as the difference of the VSWR?

Authors’ ResponseThanks for mentioning this important point. Return loss did not change in this structure. And there is not any frequency shift as well. Hence we revised our paper with the new sentence (Line 196-197):

 “The results showed that, with only 20% conductor coverage, an efficiency of 60% can be achieved while the centre frequency and return loss at the 2.45 GHz were not affected”.

3.        The author mentioned the fraying problem in an embroidery antenna structure. If the authors could briefly comment on ways to overcome this problem, it would be beneficial to future readers.

Authors’ ResponseThanks for your comment. In the revised paper we provided solution for this problem which is mentioned in the line 206-207:

“As an alternative to embroidery, commercial woven conductive fabrics which are available in the form of fabric sheet are used to realize the antenna metallic patterns stitched into the clothing, but they suffer from the same issues of degeneration of the conductive layer as well as fraying which may be overcome by shaping fabrics by means of a laser cutting machine (see Figure 4)”.

     4.      The polymer embedded antenna with PDMS fabric attachment is illustrated in Fig 5. However, it is not easy to understand the sealing effect. The authors can provide more information in Fig 5 to facilitate understanding.

Authors’ Response: Thanks for your comment. We added some explanation and the side view of the structure for Fig 5 in the revised version to make it clearer and easier to understand the sealing effect.

    5.      In line 251, the authors mentioned the fine-tuning of the PDMS relative permittivity. Are there any side effects of inclusion of other materials for tuned the relative permittivity?

Authors’ Response: Thanks for mentioning this important point. This part is moved to the line 229 in the revised paper. At the next three lines 231 to 235 we mentioned the disadvantages of the adding ceramic to PDMS and changing relative permittivity.

“The further dielectric losses, lower fringing fields due to the high permittivity which degrade the efficiency of the antenna should be considered [54]. Moreover, a larger amount of ceramic loading affects the flexibility of PDMS and complicates its fabrication process, as the mixture becomes thicker and thus harder to mix homogeneously [55]”.

    6.      In line 254, please fix the degree unit (up to 400_C)

Authors’ ResponseThanks for correcting this mistake. We fixed this mistake. (C) (Line 237 in revised paper)

     7.      If the authors added some images of the microfluidic antennas with injection alloys, it would facilitate better understanding of the concept.

Authors’ ResponseThanks for mentioning this important point. We added the Fig. 9 in page 11. This figure show the structure of 1 mm-wide microfluidic channels to construct a flexible microstrip patch antenna. In this figure the Inlets and EGaIn as a metal alloy are shown.

     8.      In line 344, why is there a 4. coming out suddenly? Is it just a typo or was something omitted? It looks like the start of another section.

Authors’ ResponseExcuse us for this mistake. It is section 4 and we fixed it and the number of other sections.

     9.      The authors can provide an example or images of the printing technique antennas on page 11 and fabrication techniques for future wearable millimeter-wave antennas on page 12-13.

Authors’ Response: Great suggestion. We added Figure 10 which shows antennas fabricated with inkjet printing in page 12 section 5. We also added Figure 11 which represents an mmW antenna and array based on etching and printing technique in the page 14 for section 7.

    10.   There are numbering issues after 5. 3D-Printed Antenna followed by 7 — fabrication techniques for future wearable millimeter-wave antennas.

Authors’ Response: Thanks for your comment. It was a mistake due to section 4 which you mentioned in the comment 8. We fixed the numbering the all of sections.

     11.   In line 441, a 2_2 microstrip patch antenna was mentioned. Is it correct with 2_2?

Authors’ ResponseThanks for your comment and very great review. Excuse us for this mistake. We meant 2×2 elements. We fixed it in revised version (Line 462).

     12.   The authors mentioned the SAR on page 13-14. There are two main human SAR measurements with body SAR and head SAR. The authors only mentioned the body SAR in the various flexible antennas. What about the head SAR? Are there any technical approaches to reduce the SARs--specific design concept or types of antennas -- without degrading its performance?

Thanks for your suggestions and comment. We agree that there are two main SAR measurements and we believe the title of Section 8 was misleading before, and apparently limited the scope of our discussion. In addition, we added some explanation in the line 526-528 section 8.

“These phantoms can be adapted for characterization of antennas-under-test (AUT) to emulate environments like human body parts (e.g., arms, torso, etc) or the head, including the skull and the brain.”

 The aim of Section 8 in the article is to introduce the reader with SAR and phantom development, and classifying the phantoms depending on their preparation states. To clarify this, we have revised the title of Section 8 to clarify better that the section deals with phantoms that can emulate both, the head as well as body parts. There are several techniques to improve the SAR in head as well as body measurements, the most common of which, include use of full ground plane to have the human-body shielded. Several of the antennas discussed in this article do utilise such approaches for reduction of SAR, and all the discussed manufacturing techniques can realise such technological solutions for reduction of SAR. Moreover, using electrically small magnetic antennas, such as a loop antenna and its variants [Q, R], modifying the antenna layout based on the antenna electric and magnetic energy density [S, T] are another solutions to improve the SAR. However, considering the scope of the article and keeping the discussion towards manufacturing techniques, we have not delved into specific details relating to the reduction of SAR.

 [Q]. M. Manteghi and A. A. Y. Ibraheem, "On the study of the near-fields of electric and magnetic small antennas in lossy media," IEEE Trans. Antennas Propag., vol. 62, no. 12, pp. 6491-6495, Dec. 2014.

[R]. R. M. Mäkinen and T. Kellomäki, "Body effects on thin single-layer slot, self-complementary, and wire antennas," IEEE Trans. Antennas Propag., vol. 62, no. 1, pp. 385-392, Jan. 2014.

[S]. X. Qing, C. K. Goh and Z. N. Chen, "A Broadband UHF Near-Field RFID Antenna," IEEE Trans. Antennas Propag., vol. 58, no. 12, pp. 3829-3838, Dec. 2010.

[T]. G. A. Casula, A. Michel, G. Montisci, P. Nepa and G. Valente, "Energy-Based Considerations for Ungrounded Wearable UHF Antenna Design," IEEE Sensors J., vol. 17, no. 3, pp. 687-694, 2017.

Reviewer 2 Report

Very interesting overview of flexible antennas in wearable devices.

It may be missing some figures representing the significant results obtained by different authors

Author Response

Comments from Reviewer 2

Very interesting overview of flexible antennas in wearable devices.

It may be missing some figures representing the significant results obtained by different authors

Authors’ Response: Thanks a lot for your nice comment and supporting our research. In the revised paper we added four figures to each section to improve our paper:

Fig. 7 to the section 3 in the page 9.

Fig. 9 to the section 4 in the page 11.

Fig. 10 to the section 5 in the page 12.

Fig. 11 to the section 7 in the page 14.

Reviewer 3 Report

This paper reviews recent advances in antennas from the aspects of materials and fabrication process. The topic is timely and should be able to attract a broad readership. Before I recommend its publication, the following issues should be addressed: 1. The copyright permission of reprint the figures should be required. 2. The manuscript contains two little figures. 3. The format of references should be checked, such as Ref.24 and the publication month is not necessary added 4. The sections of this manuscript should be numbered properly rather than 1,2,3,4,4,5,7,8,8. 5. In P10 line 344, the title in section 4 should be properly formatted. 6. This manuscript focus on the materials and fabrication technologies of flexible antennas, however, there are some latest reports on flexible-materials based antennas, such as “Carbon, 2018, 130, 164”, “Science Bulletin, 2018, 63 (9), 574”. These should also be included in the papers. 7. There are some problems in the logic of the manuscript, the author should be thoroughly checked. For example, the authors wrote that ...the fabrication process….is shown in Figure 6, after that the advantages of PDMA were described detail in P7. This made me quite confusing and I suggest that the introduction of PDMA should be put forward in section 3; The author declares the differences of this review in P3 using two paragraphs. This should be simplified. In P7 line 230-233, the meaning of the two sentences are the same.

Author Response

Comments from Reviewer 3

This paper reviews recent advances in antennas from the aspects of materials and fabrication process. The topic is timely and should be able to attract a broad readership. Before I recommend its publication, the following issues should be addressed:

1. The copyright permission of reprint the figures should be required.

Authors’ Response: Thanks for your comment. We made all of figures in our lab. All of them are ours. Just in the revised version we added Fig. 7, Fig. 10 and Fig. 11 which have permission and their official permission are added to the cover letter.

2. The manuscript contains too little figures.

Authors’ Response: Thanks for your comment. To improve our paper, in the revised version we added four figures: Fig. 7 in page 9, Fig. 9 in page 11, Fig. 10 in page 12 and Fig. 11 in page 14.

3. The format of references should be checked, such as Ref.24 and the publication month is not necessary added

Authors’ Response: Thanks for your comment. Regarding this issue we checked our references again and revised the references [1], [2], [6], [7], [8], [11], [14], [28], [29], [37], [41], [46], [48], [52], [53], [57], [60], [61], [62], [64], [65], [66], [74], [87], [90].

4. The sections of this manuscript should be numbered properly rather than 1,2,3,4,4,5,7,8,8.

Authors’ Response: Thanks for mentioning this important point. We checked and fixed this problem in revised version.

5. In P10 line 344, the title in section 4 should be properly formatted.

Authors’ Response: Excuse us for this mistake. We fixed it and the number of other sections as well.

6. This manuscript focus on the materials and fabrication technologies of flexible antennas, however, there are some latest reports on flexible-materials based antennas, such as “Carbon, 2018, 130, 164”, “Science Bulletin, 2018, 63 (9), 574”. These should also be included in the papers.

Authors’ Response: Thanks for suggesting this two papers and helping to improve our paper. In these two papers, new flexible conductive materials are introduced. Hence, we realized we should add one point more to the future research in this domain by mentioning these two references. We could open the area for researchers who are working on flexible materials (page 17, line 580-582).

Introducing new flexible materials such as graphite films [110] and graphene film (FGF) [111] as a conductive parts of antenna with a high conductivity for printing technique or new proposed fabrication techniques”.

7. There are some problems in the logic of the manuscript, the author should be thoroughly checked. For example, the authors wrote that ...the fabrication process….is shown in Figure 6, after that the advantages of PDMA were described detail in P7. This made me quite confusing and I suggest that the introduction of PDMA should be put forward in section 3;

Authors’ Response: Thanks for your comment and mentioning the important point. We changed the place of paragraph “PDMS presents advantages of high flexibility…….. electrical properties have been reported in [56],”to the fourth paragraph, line 229 of the section 3. To avoid confusion and have a reasonable flow, we placed this paragraph after introducing different polymers and conductors and embedding technique. And before the fabrication process of such polymer-embedded conductive.

8. The author declares the differences of this review in P3 using two paragraphs. This should be simplified.

Thanks for comment. We removed some repetition and tried to simplify some sentences in these two paragraphs to make it easier and clear (line 81 to 92, page 3).

A number of papers are in print providing substantial reviews on antennas for BAN systems [1427]. In [1421] the focus of the review has been heavily on the antenna designs, discussing their considerations, challenges and limitations in relation to the antennas’ operation near the human body. In [17,20], [2227], various materials and manufacturing technologies for realizing antennas for wearable applications are reviewed. However, in general they only focus on a specific class of material or fabrication technique, for instance, textile or inkjet/screen printing.

In contrast, in this paper we provide a complete survey of recent materials and fabrication methods that have been applied up to now to realize antennas for body area networks, ranging from VHF to millimeter-wave band. They include those utilized for realizing the classes of wearable antennas mentioned above, i.e., fabric-, polymer-embedded, microfluidic, and print-based antennas, as well as the emerging futuristic millimeter-wave wearable antennas. Such a review paper allows a complete view of possible methods to realize antennas for wearable applications”.

9. In P7 line 230-233, the meaning of the two sentences are the same. 

Authors’ Response: Thanks for your comment. We combined these two sentences to avoid repetition (Line 226-228).
